# Topical Cyclosporine in Oral Lichen Planus—A Series of 21 Open-Label, Biphasic, Single-Patient Observations

**DOI:** 10.3390/jcm10225454

**Published:** 2021-11-22

**Authors:** Babak Monshi, Christina Ellersdorfer, Michael Edelmayer, Gabriella Dvorak, Clemens Ganger, Christian Ulm, Klemens Rappersberger, Igor Vujic

**Affiliations:** 1Department of Dermatology and Venereology, Klinik Landstraße, Juchgasse 25, 1030 Vienna, Austria; christina.ellersdorfer@gesundheitsverbund.at (C.E.); klemens.rappersberger@gesundheitsverbund.at (K.R.); igor.vujic@gesundheitsverbund.at (I.V.); 2Clinical Division of Oral Surgery, University Clinic of Dentistry, Medical University of Vienna, Sensengasse 2a, 1090 Vienna, Austria; michael.edelmayer@meduniwien.ac.at (M.E.); clemens.ganger@meduniwien.ac.at (C.G.); christian.ulm@meduniwien.ac.at (C.U.); 3Center of Clinical Research, University Clinic of Dentistry, Medical University of Vienna, Sensengasse 2a, 1090 Vienna, Austria; 4Clinical Division of Conservative Dentistry and Periodontology, University Clinic of Dentistry, Medical University of Vienna, Sensengasse 2a, 1090 Vienna, Austria; gabriela.dvorak@meduniwien.ac.at; 5School of Medicine, Sigmund Freud Private University Vienna, Freudplatz 1, 1020 Vienna, Austria

**Keywords:** lichen planus, oral, cyclosporine, topical steroid, recalcitrant, calcineurin inhibitor

## Abstract

Topical cyclosporine (CSA) has been reported as an alternative treatment in steroid-refractory oral lichen planus (OLP), but evidence is limited and conflicting. An N-of-1 trial setting could be appropriate to evaluate interindividual differences in treatment response. We studied a series of 21 open-label, biphasic single-patient observations. Patients (15 women, 6 men) with OLP recalcitrant to topical steroids received four weeks of CSA mouth rinse (200 mg/twice daily) followed by four weeks of drug withdrawal. Pain (visual analogue scale (VAS) score), disease extent (physicians’ global assessment (PGA) score) and quality of life (Dermatology Life Quality Index (DLQI) score,) were assessed at baseline (T0), after four weeks of treatment (T1) and after another four weeks without treatment (T2). Median age was 58 years (interquartile range/IQR = 52–67) and median disease duration was 18 months (IQR = 12–44). Median baseline VAS score decreased significantly at T1 (*p* = 0.0003) and increased at T2 (*p* = 0.032) (T0 = 5 (IQR = 3–6.5); T1 = 2 (IQR = 0.5–3.4); T2 = 3 (IQR = 2–4.8)). Similarly, median baseline PGA score decreased significantly at T1 (*p* = 0.001) and increased at T2 (*p* = 0.007) (T0 = 2 (IQR = 1.3–2.5); T1 = 1 (IQR = 1–2); T2 = 2 (IQR = 1–2)). Median baseline DLQI score also decreased significantly at T1 (*p* =.027) but did not change at T2 (*p* = 0.5) (T0 = 2.5 (IQR = 1–5.8); T1 = 1 (IQR = 0–3); T2 = 1 (IQR = 1–4)). CSA responders (*n* = 16) had significantly higher median baseline VAS scores (5.2 (IQR = 5–6.5)) than nonresponders (*n* =5) (2 (IQR = 2–3.5) (*p* = 0.02). In our study, pain, disease extent and quality of life of patients with OLP improved significantly during therapy with low-dose CSA mouth rinse and exacerbated after drug withdrawal. Remarkably, patients with high initial VAS scores seemed to profit most.

## 1. Introduction

Oral lichen planus (OLP) is a chronic inflammatory disease of the oral mucosa with a reported prevalence from 0.5 to 2% [1,2]. Although sometimes asymptomatic, many patients suffer from burning sensations and impaired daily activities such as eating, drinking and talking.

The pathogenesis of OLP is not fully understood but it is widely accepted that cytotoxic CD8+ T-lymphocytes play a central role leading to lichenoid inflammation at the dermo–epidermal junction (so called interface dermatitis) with apoptosis of basal keratinocytes [1]. Extensive destruction of the basal cell layer may lead to mucosal erosions associated with severe pain and reduced quality of life [3,4]. In the course of the disease, 1–5% of patients might develop mucosal squamous cell carcinoma [5,6].

Topical corticosteroids (CS) are the mainstay of therapy in OLP, but their long-term use is limited by well-known adverse events [1,7]. Moreover, not all patients respond adequately and in some cases the disease is particularly difficult to treat. Thus, topical formulations of calcineurin inhibitors were reported as alternative therapeutic options, due to their capability to inhibit T-lymphocyte proliferation and decrease proinflammatory cytokine production [8,9,10,11,12,13,14,15,16]. In this regard, tacrolimus was recommended as the drug of first choice, while topical cyclosporine (CSA) is not part of the daily clinical routine. This might be due to limited and conflicting data on its efficacy as well as to higher costs compared to topical steroids [17,18,19,20,21].

To clarify its effect in each individual patient, we have set up a predefined, standardized treatment protocol for the use of topical CSA in patients recalcitrant to the first line therapy with triamcinolone acetonide, Volon A^®^ (Dermapharm GmbH, Vienna, Austria) since 2017. Our protocol was based on an N-of-1 trial setting in which every single patient served as her or his own control and included therapy with CSA mouth rinse for four weeks followed by subsequent discontinuation for another four weeks to evaluate individual changes in pain, clinical picture and quality of life. The aim of the current study was to analyze the efficacy of topical CSA in steroid-refractory OLP in this standardized single-patient setting.

## 2. Materials and Methods

The study was approved by the ethics committee of the City of Vienna (EK 16–188-VK) and describes a series of 21 open-label, biphasic single-patient observations. We included all consecutive patients with symptomatic OLP from January 2017 to January 2019 who fulfilled inclusion criteria and were willing to participate in the study (Figure 1, Table 1).

Patients were diagnosed and treated at the interdisciplinary outpatient clinic for oral mucosal diseases, a cooperation of the Department for Dermatology, Klinik Landstraße and the University Clinic of Dentistry, Vienna, Austria. OLP was classified according to the predominant oral lesions into reticular, atrophic/erythematous or erosive/ulcerative subtype as described previously [22]. Other oral inflammatory diseases such as lichenoid contact mucositis or lichenoid drug reactions were excluded through patients’ medication history and joint clinical examination by dentists and dermatologists. Autoimmune blistering diseases, such as mucous membrane pemphigoid, were excluded by histology, direct and indirect immunofluorescence studies as well as enzyme-linked immunosorbent assay (ELISA; MBL, Nagoya, Japan) testing. Patients were instructed to rinse their mouth with 2 mL CSA (100 mg/mL) twice daily for 5 min and, after spitting out the residual liquid, to refrain from eating and drinking for 30 min. Therapy was continued for four weeks and subsequently suspended for another four weeks. During this protocol, patients had to have three visits, i.e.: (i) baseline visit at the start of CSA treatment (T0); (ii) after four weeks of topical CSA at which therapy was withdrawn (T1), and (iii) after another four weeks without any treatment (T2). At each visit patients were assessed by two specialists for oral medicine (I.V. and/or B.M.) regarding: (i) pain; (ii) disease extent and (iii) quality of life. Pain was measured using a visual analogue scale (VAS) (0–10 points, 0 = no pain; 10 = the worst pain possible; counted in intervals of 0.5) [23]. To obtain a measurement of disease extent we graded patients’ oral lesions using a physicians’ global assessment (PGA) score ranging from 0–4 points (0 = no clinical signs; 4 = extensive erosive/ulcerative disease; counted in intervals of 0.5). The impact on quality of life was assessed using the Dermatology Life Quality Index (DLQI) questionnaire. This tool is widely used in dermatological conditions and scores range from 0 to 30 points (the higher the score, the greater the impairment) and is available as supplementary file [24]. For further statistical analyses, patients were classified as responders if they experienced a reduction in their VAS scores from T0 to T1 and as nonresponders if they experienced no reduction or an increase in their VAS scores from T0 to T1. The rationale to choose the validated, comparable and widely used VAS score as the main outcome variable was based on the assumption that OLP pain, but not the clinical picture, significantly influences quality of life of patients and is the main clinical feature to guide therapeutic decisions. Furthermore, OLP-specific therapy might influence pain while the clinical disease extent does not respond in the same way [21,25]. After four weeks of treatment, CSA serum concentrations were measured 2–4 h after the last application using Roche Elecsys^®^ (Roche Diagnostics Limited, Burgess Hill, UK) Cyclosporine Assay (detection level > 30 ng/mL).

Median and interquartile ranges (IQR) were used to describe continuous variables and absolute and relative frequencies were used to describe categorical variables. The two-sided Wilcoxon signed-rank test was used to compare paired measurements and the Wilcoxon rank-sum test was used to compare independent measurements. To estimate correlations between variables we used Pearson’s correlation coefficient (*r*). All computations were performed with SPSS (IBM^®^ SPSS Statistics, New York, NY, USA). *p*-values < 0.05 were considered as statistically significant. The datasets used and analyzed during the current study are available from the corresponding author on reasonable request.

## 3. Results

We included 21 patients in the study and their clinical characteristics are depicted in Table 2.

Three patients, two women and one man, suffered from concomitant involvement of the genital mucosa (14.3%) and two patients, both women, had additional lichen planus of the skin (9.5%). All patients had been treated previously with topical CS (triamcinolone acetonide, Volon A^®^ (Dermapharm GmbH, Vienna, Austria) for at least four weeks and one patient had been treated with additional systemic CS (Appendix A).

At baseline (T0) VAS and PGA scores correlated significantly (r = 0.562; *p* = 0.008; Pearson’s correlation coefficient; *n* = 21) while no correlation was found between baseline DLQI and VAS scores (r = 0.36; *p* = 0.15; *n* = 17) or PGA scores (r = 0.1; *p* = 0.7; *n* = 17), respectively.

### 3.1. Pain (VAS Scores)

At T0 (baseline), the median VAS score was 5 (IQR = 3–6.5) (*n* = 21).

At T1 (first follow-up after four weeks of continuous CSA treatment), median VAS scores decreased significantly to 2 (IQR = 0.5–3.4) (*p* = 0.0003, Wilcoxon signed-rank test; *n* = 21). In detail, median VAS scores decreased in 16 patients (76%) (3.5, IQR = 2.4–5)*,* increased in 3 (14%) (1, IQR = 0.8–1)*,* and remained unchanged in 2 patients (10%).

At T2 (four weeks after discontinuation of CSA and the end of observation period), median VAS scores increased significantly to 3 (IQR = 2–4.8) compared to T1 (*p* = 0.032; *n* = 21). In detail, median VAS scores increased in 13 patients (62%) (2.5, IQR = 1.5–3.5), decreased further in 5 (24%) (2, IQR = 1.5–2), and remained unchanged in 3 patients (14%).

Over the whole study period, pain decreased significantly from T0 to T2 (*p* = 0.023), and detailed analysis revealed that median VAS scores decreased in 14 patients (67%) (2.5, IQR = 1.8–4.5), increased in 4 (19%) (2.5, IQR = 1.8–3.1) and remained unchanged in 3 patients (14%) (Figure 2, Figure 3 and Appendix A).

### 3.2. Disease Extent (PGA Scores)

At T0, the median PGA score was 2 (IQR = 1.3–2.5) (*n* = 21).

At T1, median PGA scores decreased significantly to 1 (IQR = 1–2) (*p* = 0.001; *n* = 21). In detail, median PGA scores decreased in 14 (67%) (1, IQR = 1–1) and remained unchanged in 7 patients (33%).

At T2, median PGA scores increased significantly to 2 (IQR = 1–2) compared to T1 (*p* = 0.007; *n* = 21). In detail, median PGA scores increased in 11 (52%) (1, IQR = 1–1), decreased in 2 (10%) (0.75, IQR = 0.6–0.9) and remained unchanged in 8 patients (38%).

Over the whole study period, the extent of the disease did not change significantly from T0 to T2 (*p* = 0.11), and detailed analysis revealed that median PGA scores decreased in 10 (48%) (0.75, IQR = 0.5–1), increased in 3 (14%) (1, IQR = 0.75–1) and remained unchanged in 8 patients (38%) (Figure 2). Two clinical examples of patient two and patient eight are depicted in Figure 4.

### 3.3. Quality of Life (DLQI Scores)

At T0, the median DLQI score was 2.5 (IQR = 1–5.8) (*n* = 17).

At T1, median DLQI scores decreased significantly to 1 (IQR = 0–3) (*p* = 0.027; *n* = 17). In detail, median DLQI scores decreased in 12 (71%) (2.5, IQR = 1.8–4.5) and increased in 4 (24%) (1, IQR = 1–2.8) and remained unchanged in 1 patient (6%).

At T2, median DLQI scores did not change significantly compared to T1 (1, IQR = 1–4; *p* = 0.5; *n* = 14). In detail, median DLQI scores increased in seven (50%) (1, IQR = 1–1.5), decreased in four (29%) (1.5, IQR = 1–2.3) and remained unchanged in three patients (21%).

Over the whole study period, quality of life did not change significantly from T0 to T2 (*p* = 0.06, *n* = 14) and detailed analysis revealed that DLQI scores decreased in eight (58%) (3, IQR = 2.5–5), increased in three (21%) (2, IQR = 1.5–2.5) and remained unchanged in three patients (21%) (Figure 2).

We compared patients who responded to topical CSA treatment (*n* = 16) with those who did not respond (*n* = 5), as defined by reduction in VAS scores from T0 to T1, using the Wilcoxon rank-sum test. Responders had significantly higher VAS scores at baseline (5.2, IQR = 5–6.5) than nonresponders (2; IQR = 2–3.5) (*p* = 0.02), but did not differ regarding baseline PGA, DLQI scores or other characteristics such as age and sex (Table 3). Additional analysis on whether primary responders showed higher increases of VAS, PGA and DLQI scores compared to nonresponders from T1 to T2, four weeks after discontinuation of CSA revealed no significant differences between the two groups (Appendix A).

From 21 patients, only 2 had detectable serum CSA levels at week four (T1) (40.5 ng/mL and 48.75 ng/mL) and no patients had detectable CSA levels at week eight (T2) (<30 ng/mL).

## 4. Discussion

OLP is a debilitating mucosal disease which can be unresponsive to topical CS and might run a chronic course [26]. In this study, we report the outcomes of a series of 21 consecutive patients with steroid-refractory OLP, who underwent a predefined treatment regimen with low-dose CSA mouth rinse (2 mL twice daily) for four weeks followed by discontinuation of treatment for another four weeks. Pain (VAS), clinical picture (PGA) and quality of life (DLQI) were assessed at the beginning of treatment, at four weeks—when CSA therapy was discontinued—and after another four weeks without therapy. Overall, we found that four weeks of continuous topical CSA resulted in a significant reduction in pain (*p* = 0.0003), and vice versa, discontinuation of CSA led to a significant and relatively swift recurrence of pain at week eight (*p* = 0.032) (Figure 2, Figure 3 and Appendix A).

In this regard, it needs to be mentioned that the existing literature on the efficacy of CSA in OLP is inconsistent. While some studies have demonstrated clear benefits of the compound compared to placebo, others have failed to do so [8,12,27,28,29,30,31]. In addition, and further causing ambiguity, single trials, a systematic review and a meta-analysis reported that the efficacy of CSA was similar to topical CS such as triamcinolone acetonide [7,17,21,32,33,34]. In this recent meta-analysis, it was also stressed that additional studies are needed to further clarify the role of CSA in OLP, not at least in the light of its higher costs compared to the standard therapy with topical CS [17]. One explanation for the aforementioned discrepancies between different studies might be the fact that the average efficacy of a compound such as CSA, found in RCTs, may not reflect its effect on an individual level, a problem known as heterogeneity of treatment effects [35,36]. Poor generalizability of RCTs’ results is particularly likely in rare conditions such as OLP, in which, for example, adequate recruitment of homogenous study-groups may be difficult. In such a context, N-of-1 trial designs, defined as multiple-period crossover experiments comparing two treatments within individual patients, might be superior to conventional RCTs [36]. Although our study does not fulfill all those criteria, its standardized protocol is similar to an N-of-1 trial set-up, as it compares each single patients’ response separately, and every patient served as her or his own control. Thus, it allowed us to analyze interindividual differences in treatment responses. Through this specific approach we found that CSA therapy was effective in 16 out of 21 patients. Eleven of those were swift responders with an initial reduction in their median baseline VAS scores from 5 (IQR = 4–6.5) to 1 (IQR = 0.3–1.5) at week four (T1) and on the other hand an immediate increase in pain to median VAS scores of 4 (IQR = 2.5–4.3) after discontinuation of treatment (T2). Another five patients with initial median VAS scores of 6 (IQR = 5–6.5) showed a significant and lasting reduction in pain to 3 (IQR = 2–4) at week four and 2 (IQR = 0–3) at week eight, respectively, which might indicate a prolonged treatment effect. Interestingly, further analysis revealed that patients with high initial VAS scores seemed to profit most from CSA therapy (*p* = 0.02) (Table 3). In contrast, five patients were nonresponders, showing either no significant changes or even an increase in their median VAS scores from 2 (IQR = 2–3.5), to 2.5 (IQR = 2–3.5) at T1 and to 5 (IQR = 3–5) at T2, respectively (Figure 2, Figure 3 and Appendix A).

Mirroring VAS scores across the study period, the clinical picture as measured by PGA improved significantly after four weeks of CSA treatment (*p* = 0.001) and deteriorated after four weeks without therapy (*p* = 0.007) (Figure 2 and Figure 4). Therefore, both, patient-dependent pain scores as well as investigator-dependent PGA scores indicate that overall, CSA mouth rinse might be effective in patients with steroid-refractory OLP. However, treatment effects might be heterogeneous, as most patients who respond to initial therapy relapsed after drug withdrawal without signs of lasting effects, while a few patients experienced continuous remissions. On the other hand, it seems that in a minority of patients, topical CSA might have no adequate effect at all.

Mucosal diseases have an impact on basic daily activities such as eating, drinking and talking, and to date, quality of life has not been assessed in therapeutic trials of OLP. In our study, DLQI scores roughly matched the course of VAS scores, as they decreased significantly after treatment with CSA at week four (*p* = 0.027) and increased after withdrawal of CSA, although not statistically significantly (*p* = 0.5). The latter nonsignificant result might be due to missing DLQI scores which were not available in four patients at T0/T1 and in seven patients at T2. However, our study clearly points out that quality of life constitutes an integral part in assessing OLP patients and should therefore be part of future studies.

We recorded no adverse events, apart from burning sensations during topical application of CSA in four patients, and analysis of serum concentrations showed that only two patients had detectable CSA levels, which were below 50 ng/mL and deemed to be insufficient to explain our clinical observations.

Some limitations of our study warrant further discussion. First, OLP is a chronic, immune-mediated condition that tends to wax and wane, which might have contributed to the heterogeneity of our study‘s results. Secondly, although VAS scores and DLQI questionnaires were completed by patients separately from investigators, we cannot exclude an observational bias as the study was neither blinded nor randomized. Thirdly, we cannot rule out a selection bias as patients were recruited in a specialized outpatient clinic at a tertiary referral center. To assess disease severity, we used a nonvalidated physician’s global assessment (PGA), based on published clinical criteria and conducted by two experienced dermatologists (I.V. and/or B.M.) [22]. In this regard, it needs to be mentioned that different scoring systems used to measure the extent of inflammatory oral diseases have been proposed, but to date no single, generally available tool, similar to the psoriasis area and severity index (PASI) for psoriasis, has been established [33,37]. To address this issue, we conducted correlation studies and found a significant association of PGA and VAS scores (*p* = 0.008), indicating that PGA truly reflects the disease burden in our patients. Finally, although DLQI is a validated and commonly used tool to measure the social impact of dermatological conditions, other tools, such as the Chronic Oral Mucosal Diseases Questionnaire (COMDQ), the Oral Health Impact Profile (OHIP) or the Reticular-Erythematous-Ulcerative (REU) scoring system, might have yielded a more differentiated picture in the setting of OLP [38,39,40].

## 5. Conclusions

In conclusion, we demonstrate that low-dose CSA mouth rinse (200 mg twice daily) might be effective in patients with OLP recalcitrant to topical CS. In this series of 21 biphasic, single-patient observations, pain, disease extent and quality of life improved significantly under topical CSA, which was well tolerated. We noticed heterogeneity in individual treatment responses as patients with high initial VAS scores seemed to profit most, while a subgroup of patients did not benefit at all. Therefore, N-of-1 trial designs should be considered in future studies which are needed to further detail the role of CSA in the care of patients with symptomatic OLP.

## Figures and Tables

**Figure 1 jcm-10-05454-f001:**
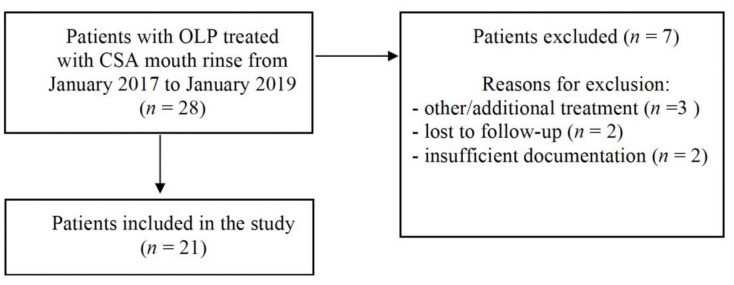
Flow chart of inclusion process. OLP: oral lichen planus; CSA: cyclosporine.

**Figure 2 jcm-10-05454-f002:**
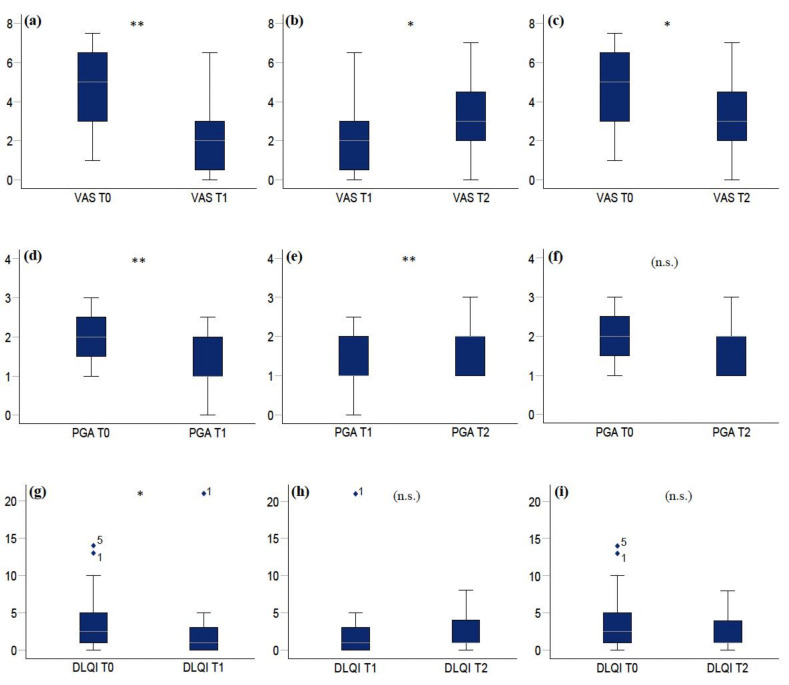
Boxplots depicting comparisons of VAS (visual analogue scale), PGA (physicians’ global assessment) and DLQI (Dermatology Life Quality Index) scores before CSA treatment (T0), after four weeks of treatment (T1) and after four weeks without treatment (T2). Upper row (**a**–**c**) VAS score: (**a**) T0/T1, (**b**) T1/T2, (**c**) T0/T2; middle row (**d**–**f**) PGA score: (**d**) T0/T1, (**e**) T1/T2, (**f**) T0/T2; lower row (**g**–**i**) DLQI score: (**g**) T0/T1, (**h**) T1/T2, (**i**) T0/T2. * *p* ≤ 0.05, ** *p* ≤ 0.01, (n.s.): nonsignificant.

**Figure 3 jcm-10-05454-f003:**
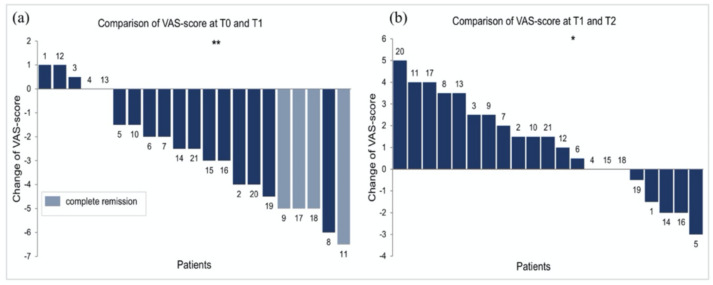
Waterfall plots depicting individual patients’ changes of VAS scores. Numbers 1–21 indicate individual patients: (**a**) VAS score changes from T0 (baseline) to T1 (after four weeks of CSA). An increase in VAS scores was found in 3 (patients 1, 12, 3), no change in 2 (patients 4, 13) and a reduction in VAS scores in 16 patients (*p* = 0.0003). Four patients were free of pain after four weeks of topical CSA (light blue). (**b**) VAS score changes from T1 to T2 (after four weeks without treatment). An increase in VAS scores was found in 13 patients, no change in 3 (patients 4, 15, 18) and a further reduction in VAS scores in 5 (patients 19, 1, 14, 16, 5) (*p* = 0.032). * *p* ≤ 0.05, ** *p* ≤ 0.01.

**Figure 4 jcm-10-05454-f004:**
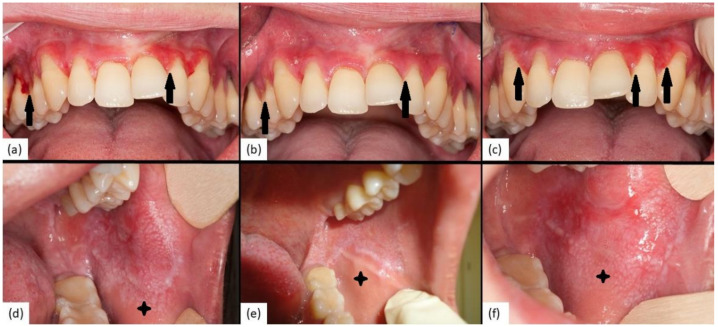
Clinical pictures of patient 2 (panel (**a**–**c**)) and patient 8 (panel (**d**–**f**)) at baseline (T0), after 4 weeks of CSA (T1) and after another 4 weeks without therapy (T2). PGA of patient 2 was 2 at T0 (**a**), 0.5 at T1 (**b**) and 1 at T2 (**c**) (arrows indicate affected and improved areas). PGA of patient 8 was 3 at T0 (**d**), 1 at T1 (**e**) and 2.5 at T2 (**f**) (asterisk indicates affected and improved area).

**Table 1 jcm-10-05454-t001:** Inclusion and exclusion criteria.

Inclusion Criteria	Exclusion Criteria
Clinically and histologically confirmed diagnosis of OLP	Incomplete documentation
No prior treatments for OLP for at least two weeks before start of topical CSA	Treatment with other topical and/or systemic compounds in the two weeks before and/or during the study period
Treatment with topical CSA (2 mL twice daily) for four weeks	
After four weeks of CSA treatment, drug withdrawal for at least four weeks	
Baseline visit (T0), 1st follow-up (week 4; T1) and 2nd follow-up (week 8; T2)	Missed follow-up visits

**Table 2 jcm-10-05454-t002:** Baseline characteristics of all 21 patients. IQR: interquartile range; OLP: oral lichen planus.

Patients (*n* = 21)	Total Number (%)
Female Male	15 (71) 6 (29)
Median age	Years (IQR)
All patients Female Male	58 (52–67) 58 (52–66) 59 (58–70)
Duration until diagnosis of OLP	Months (IQR)
Median duration	18 (12–44)
Localisation of lesions	Number of patients
Gingiva	7
Buccal muosa	2
Buccal mucosa/Gingiva	6
Buccal mucosa/Tongue	3
Buccal mucosa/Tongue/Lips	2
Tongue/Lips	1
Clinical subtype of OLP	Number of patients
Erosive/ulcerative	15
Erythematous/atrophic	3
Reticular	3
Involvement of other parts	Number of patients (%)
Genital involvement (%)	3 (14)
Cutaneous involvement (%)	2 (10)

**Table 3 jcm-10-05454-t003:** Wilcoxon rank-sum test comparing patients who initially responded to topical CSA treatment (responders; *n* = 16) with those who did not respond (nonresponders; *n* = 5). Responders had significantly higher baseline VAS scores than nonresponders (bold number), but groups did not differ significantly regarding other characteristics. * Evaluation of baseline DLQI scores was based on 17 patients.

Variable	Overall (*n* = 21) No. (%) Median (Q1–Q3)	Responder (*n* = 16) No. (%) Median (Q1–Q3)	Nonresponder (*n* = 5) No. (%) Median (Q1–Q3)	*p*-Value
Sex				0.26
Male	6 (29%)	6 (38%)	0 (0%)	
Female	15 (71%)	10 (625)	5 (100%)	
Age	58 (52–66)	59 (51–70)	58 (52–58)	0.50
VAS T0	5 (3–6.5)	5.2 (5–6.5)	2 (2–3.5)	0.02
PGA T0	2 (1.5–2.5)	2 (1.9–2.6)	2 (1–2)	0.18
DLQI T0 *	2.5 (1–5)	3 (1–5)	2 (1–10)	0.84

## Data Availability

The data in this study are available on request from the corresponding author. The data are not publicly available due to privacy issues.

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
