# Peer review of "Topical Cyclosporine in Oral Lichen Planus—A Series of 21 Open-Label, Biphasic, Single-Patient Observations"

_jcm, 2021, doi:10.3390/jcm10225454_

Round 1

Reviewer 1 Report

Dear editor

Thanks for giving me the opportunity to review the paper of Monshi et al., which is titled as “Topical Cyclosporin in Oral Lichen Planus – A Series of 21 Open-label, Biphasic Single-patient Observations”.

Overall the paper is structured well and moving between sections are smooth, except for the abstract.  However, the followings are some comments that could help authors improve the manuscript. Minor English auditing is required to check punctuations.

Abstract

Unlike other sections in this paper, the abstract is a bit confusing and needs to be restructured thoroughly

  • line 14 to line 16: this sentence is a bit wordy and causes confusion, I had to read it several times to link the ideas. I would suggest rephrasing and shortening this section.
  • line 18 and 19: please add what does VAS, PGA, and DLQI mean.
  • line 21: please add "was" before 18
  • line 22: again, this section is very wordy. Readers need to understand the abstract smoothly and easily. In the current version, It would take time to link these numbers with each other. I would suggest rephrasing in a different style.

Introduction

  • line 38 to 40: please add a reference to support this statement
  • line 42: add "," after "disease"

Methodology

  • line 62: I am wondering how to did you get ethical approval to conduct a clinical trial from the city of Vienna. Is there a collaboration between the city council and other official health departments to grant this kind of approval? Why you did not get it from the university clinic of dentistry?
  • Table 1: please mention which OLP diagnostic criteria that you used to confirm the OLP diagnosis
  • line 85: Although I understand these letters (IV and BM) are referred to specific authors, it would be a bit confusing for readers at the beginning and they would believe that IV and BM are kinds of assessments. I would suggest providing simple clarification by saying for example "all patients were assessed by 2 oral medicine specialists (IV and or BM)" or so on.
  • As you used three indices to measure different variables (VAS, PGA, DLQI), please include more detailed descriptions of these indices in the supplementary file. Please cite the authors/developers/validators of these indices.
  • Please provide a general description or definition for (N-of-1 trial design).
  • Up to my knowledge, it is not usual to refer to significance values like α=5%, I would suggest using the common format, p <0.05.

Results

Table 2: regarding the lesions localization, the total number of patients is 37, although there were 21 patients so apparently there were patients with multiple OLP lesions. Please clarify this point so at the end the total number of patients should be 21.

Author Response

REVIEWER I.

We thank reviewer 1 for her or his astute comments and have detailed our amendments according to her or his suggestions (in italics).

Abstract

  • line 14 to line 16: this sentence is a bit wordy and causes confusion, I had to read it several times to link the ideas. I would suggest rephrasing and shortening this section.

We agree with reviewer 1 and have rephrased the paragraph as suggested:

An N-of-1 trial setting could be appropriate to evaluate inter-individual differences in treatment response. We studied a series of 21 open-label, biphasic single-patient observations.

  • line 18 and 19: please add what does VAS, PGA, and DLQI mean.

As suggested, we have added the meanings of the abbreviations to the abstract in parenthesis.

Pain (visual analogue scale [VAS]-score), disease extent (physicians´ global assessment [PGA]-score) and quality of life (Dermatology Life Quality Index [DLQI]–score,) were assessed at baseline(T0), after four weeks of treatment(T1) and after another four weeks without treatment(T2).

  • line 21: please add "was" before 18

As suggested we have added was to the sentence.

Median age was 58 years (interquartile range/IQR=52–67) and median disease duration was 18 months (IQR=12–44).

  • line 22: again, this section is very wordy. Readers need to understand the abstract smoothly and easily. In the current version, It would take time to link these numbers with each other. I would suggest rephrasing in a different style.

Again we completely agree with reviewer 1 and we have amended the paragraph so readers might find it easier to understand the results in the abstract. However, we were reluctant to omit the data, as we believe this forms an integral part of the abstract.

Median baseline VAS-score decreased significantly at T1(p=.0003) and increased at T2(p=.032) (T0=5[IQR=3–6.5];T1=2[IQR=0.5–3.4];T2=3[IQR=2-4.8]). Similarly, median baseline PGA-score decreased significantly at T1(p=.001) and increased at T2(p=.007) (T0=2[IQR=1.3–2.5];T1=1[IQR=1–2];T2=2[IQR=1–2]). Median baseline DLQI-score also decreased significantly at T1(p =.027), but did not change at T2(p =.5) (T0=2.5[IQR=1–5.8];T1=1[IQR=0–3];T2=1[IQR=1–4]). CSA-responders (n=16) had significantly higher median baseline VAS-scores (5.2[IQR=5–6.5]) than non-responders (n =5) (2[IQR=2–3.5](p =.02).

Introduction

  • line 38 to 40: please add a reference to support this statement

As suggested we have added an appropriate reference to this statement.

The pathogenesis of OLP is not fully understood but it is widely accepted that cytotoxic CD8+ T-lymphocytes play a central role leading to lichenoid inflammation at the dermo-epidermal junction (so called interface dermatitis) with apoptosis of basal keratinocytes [1].

  • line 42: add "," after "disease"

As suggested we have added „,“ and reviewed our manuscript for spelling- and comma-mistakes.

In the course of the disease, 1-5% of patients might develop mucosal squamous cell carcinoma [5,6].

Methodology

  • line 62: I am wondering how to did you get ethical approval to conduct a clinical trial from the city of Vienna. Is there a collaboration between the city council and other official health departments to grant this kind of approval? Why you did not get it from the university clinic of dentistry?

The appropriate ethics committee for studies, which are conducted at any hospital of the “Wiener Gesundheitsverbund”, is the ethics committee of the city of Vienna/AUSTRIA. Although this study was a collaboration between the University Clinic of Dentistry/Vienna and the Department of Dermatology, Klinik Landstraße, Wiener Gesundheitsverbund, Vienna, it took mainly place at the interdisciplinary outpatient clinic for oral mucosal diseases, at the Dermatology Department. Therefore, we obtained the approval as stated.

  • Table 1: please mention which OLP diagnostic criteria that you used to confirm the OLP diagnosis

We have detailed which clinical and histological criteria we used for diagnosis of OLP in the methods section of our manuscript. As this is a lengthy paragraph we are reluctant to add these criteria to table 1. However, if Reviewer 1 insists, we will be happy to adjust the table or the table legend accordingly.

  • line 85: Although I understand these letters (IV and BM) are referred to specific authors, it would be a bit confusing for readers at the beginning and they would believe that IV and BM are kinds of assessments. I would suggest providing simple clarification by saying for example "all patients were assessed by 2 oral medicine specialists (IV and or BM)" or so on.

We thank reviewer 1 for this comment and agree with her/him. We therefore amended the sentence as suggested.

At each visit patients were assessed by two specialists for oral medicine (I.V. and/or B.M.) regarding (i) pain, using a visual analogue scale (VAS) (0–10 points, 0=no pain; 10=the worst pain possible; counted in intervals of .5) [23], (ii) disease extent using a physicians´ global assessment (PGA) score (0–4 points; 0=no clinical signs; 4=extensive erosive disease; counted in intervals of .5) and (iii) quality of life using the Dermatology Life Quality Index (DLQI) questionnaire (0-30 points; the higher the score, the greater the impairment) [24].

  • As you used three indices to measure different variables (VAS, PGA, DLQI), please include more detailed descriptions of these indices in the supplementary file. Please cite the authors/developers/validators of these indices.

As suggested we have included the DLQI questionnaire as a supplement and have referenced this in the text and references, accordingly. We have referrenced VAS-scoring system in the text (reference 23). We have also detailed in our manuscript how VAS-scores were evaluated with patients (patients graded their pain from 0-10 in .5 intervalls). PGA is a fairly subjective method (still very reproducable) to assess disease activity in patients with OLP and is also commonly used for studies of other dermatological conditions like psoriasis. We tried to meet potential concerns by discussing this in the limitations section of our manuscript. We have amended all relevant sections and hope that all points on this issue have now been addressed satisfactorily. 

  • Please provide a general description or definition for (N-of-1 trial design).

As suggested we have added a general definition of N-of-1 trial design to the discussion section.

In such a context, N-of-1 trial designs, defined as multiple-period crossover experiments comparing two treatments within individual patients, might be superior to conventional RCTs.  

  • Up to my knowledge, it is not usual to refer to significance values like α=5%, I would suggest using the common format, p <0.05.

We agree with reviewer 1 and have amended this sentence according to her or his suggestion.

p-values <.05 were considered as statistically significant.

Results

Table 2: regarding the lesions localization, the total number of patients is 37, although there were 21 patients so apparently there were patients with multiple OLP lesions. Please clarify this point so at the end the total number of patients should be 21.

The numbers were higher than 21 because many patients had lesions on more than one location, however we have amended Table 2 according to the suggestions made by Reviewer 1.

Patients(n = 21)

Total number (%)

   Female

   Male

   15 (71)

   6 (29)

Median age

Years (IQR)

   All patients

   Female

   Male

   58 (52–67)

   58 (52–66)

   59 (58–70)

Duration until diagnosis of OLP

Months (IQR)

   Median duration

   18 (12–44)

Localisation of lesions

Number of patients

   Gingiva

   Buccal mucosa

   Buccal mucosa/Gingiva

   Buccal mucosa/Tongue

   Buccal mucosa/Tongue/Lips

   Tongue/Lips

   7

   2

   6

   3

   2

   1

Clinical subtype of OLP

Number of patients

   Erosive/ulcerative

   Erythematous/atrophic

   Reticular

   15

   3

   3

Involvement of other parts

Number of patients (%)

   Genital involvement (%)

   Cutaneous involvement (%)

   3 (14)

   2 (10)

Reviewer 2 Report

The text is well-written and the results well presented. 

It is another study that provides evidence that topical calcineurin inhibitors can be of some use in Oral lichen planus.  It is a more expensive treatment and we do not have enough evidence of long term safety regarding the risk of olp malignant transformation. 

Yet this paper is a hood source for evidence based Medicine.

Author Response

REVIEWER II.

The text is well-written and the results well presented. 

It is another study that provides evidence that topical calcineurin inhibitors can be of some use in Oral lichen planus.  It is a more expensive treatment and we do not have enough evidence of long term safety regarding the risk of olp malignant transformation. 

Yet this paper is a hood source for evidence based Medicine.

We are very grateful for the reviewer´s comments and very happy that she or he does not seem to have any objections against publishing our study in the JCM.

Reviewer 3 Report

The article is well written. The study is well designed, with clear inclusion and exclusion criteria to rule out other causes of lichenoid inflammation or other vesiculobullous disease. I also find it helpful to know if serum levels of CSA were present after a 5 minute swish and spit of an oral solution. I appreciate the presentation of data in Figures 2 and 3.

Minor revisions recommended:

  1. There is inconsistent use of the spelling "cyclosporine" (line 50) vs. "cyclosporin" (title, line 13, etc.). In the United States and Canada, we use "cyclosporine."
  2. I do not agree with the classifications of OLP as "erosive" vs. erythematous/atrophic vs. reticular. I prefer the term ulcerative (a break in the epithelium which clinically presents with a yellow pseudomembrane) vs. erythematous/atrophic vs. reticular. I find clinicians often confuse the terms erosive vs. erythematous. Please consider revising.
  3. I do not think this can be modified at this point, but I prefer the use of the "REU" scale for scoring oral lichen planus or lichenoid diseases, such as chronic GVHD. In this way, severity can be scored based on the presence and extent of reticulation, erosions, and ulceration. I do appreciate the clinical photographs with accompanying PGA scoring in Figure 4 for clarification though.
  4. I would add in the discussion that oral lichen planus is a chronic, immune-mediated condition that can be idiopathic or secondary to systemic disease or medications, and it naturally tends to wax and wane, which may have contributed to the heterogeneity of the treatment results (e.g. some patients did not respond, some patients had longer lasting results).
  5. On a stylistic note, while the paper is well written, I do think additional commas are needed in between phrases and lists. For example, on lines 207 - 210: "While some studies have demonstrated clear benefits of the compound compared to placebo others failed to do so. In addition and further causing ambiguity, single trials, a systematic review and a meta-analysis reported that the  efficacy of CSA was similar to topical CS such as triamcinolone acetonide." I would rewrite as: "While some studies have demonstrated clear benefits of the compound compared to placebo, others failed to do so. In addition, and further causing ambiguity, single trials, a systematic review, and a meta-analysis reported that the efficacy of CSA was similar to topical CS such as triamcinolone acetonide."

Author Response

REVIEWER III.

We thank reviewer 3 for her or his astute comments and have detailed our amendments according to her or his suggestions (in italics).

  1. There is inconsistent use of the spelling "cyclosporine" (line 50) vs. "cyclosporin" (title, line 13, etc.). In the United States and Canada, we use "cyclosporine."

We completely agree, and have amended cyclosporin to cyclosporine.

  1. I do not agree with the classifications of OLP as "erosive" vs. erythematous/atrophic vs. reticular. I prefer the term ulcerative (a break in the epithelium which clinically presents with a yellow pseudomembrane) vs. erythematous/atrophic vs. reticular. I find clinicians often confuse the terms erosive vs. erythematous. Please consider revising.

Indeed, we completely agree with reviewer 3 that for the unwary clinician erythematous forms of mucosal diseases, including OLP, might resemble erosions. However, we believe the term erosive is more accurate, as this denotes superficial detachment of epithelium from the underlying connective tissue (in OLP due to massive interface changes). In our experience, ulcerative processes are less frequent in OLP and involve deeper parts of the lamina propria and the subepithelial tissues. We therefore amended, in accordance with the reviewer´s comment, erosive to erosive/ulcerative.

  1. I do not think this can be modified at this point, but I prefer the use of the "REU" scale for scoring oral lichen planus or lichenoid diseases, such as chronic GVHD. In this way, severity can be scored based on the presence and extent of reticulation, erosions, and ulceration. I do appreciate the clinical photographs with accompanying PGA scoring in Figure 4 for clarification though.

We thank reviewer 3 for her or his commendation on the clinical photographs and PGA scoring. Indeed, we are not able to change the scoring system of our study at this stage, but have included REU score in the limitations section of our manuscript.

  1. I would add in the discussion that oral lichen planus is a chronic, immune-mediated condition that can be idiopathic or secondary to systemic disease or medications, and it naturally tends to wax and wane, which may have contributed to the heterogeneity of the treatment results (e.g. some patients did not respond, some patients had longer lasting results).

As suggested we have amended our manuscript and have added the paragraph to the limitations section of the discussion.

Some limitations of our study warrant further discussion. First, OLP is a chronic, immune-mediated condition that tends to wax and wane, which might have contributed to the heterogeneity of our study`s results.

  1. On a stylistic note, while the paper is well written, I do think additional commas are needed in between phrases and lists. For example, on lines 207 - 210: "While some studies have demonstrated clear benefits of the compound compared to placebo others failed to do so. In addition and further causing ambiguity, single trials, a systematic review and a meta-analysis reported that the  efficacy of CSA was similar to topical CS such as triamcinolone acetonide." I would rewrite as: "While some studies have demonstrated clear benefits of the compound compared to placebo, others failed to do so. In addition, and further causing ambiguity, single trials, a systematic review, and a meta-analysis reported that the efficacy of CSA was similar to topical CS such as triamcinolone acetonide."

As suggested we have added „,“ and reviewed our manuscript for spelling- and comma-mistakes.
